# Contaminant occurrence, distribution and ecological risk assessment of phthalate esters in the Persian Gulf

**Maria Khishdost[1,2], Sina Dobaradaran[3,4,5], Gholamreza Goudarzi[1,2,6], Afshin Takdastan[1,2], Ali Akbar Babaei[1,2]\***

1 Department of Environmental Health Engineering, School of Public Health, Ahvaz Jundishapur University of Medical Sciences, Ahvaz, Iran, 2 Environmental Technologies Research Center, Ahvaz Jundishapur University of Medical Sciences, Ahvaz, Iran, 3 Systems Environmental Health and Energy Research Center, The Persian Gulf Biomedical Sciences Research Institute, Bushehr University of Medical Sciences, Bushehr, Iran, 4 Department of Environmental Health Engineering, Faculty of Health and Nutrition, Bushehr University of Medical Sciences, Bushehr, Iran, 5 Instrumental Analytical Chemistry and Centre for Water and Environmental Research (ZWU), Faculty of Chemistry, University of Duisburg-Essen, Essen, Germany, 6 Air Pollution and Respiratory Diseases Research Center, Ahvaz Jundishapur University of Medical Sciences, Ahvaz, Iran

\* ababaei52@gmail.com

**Data Availability Statement:** All relevant data are within the paper.

**Funding:** We thank Ahvaz Jundishapur University of Medical Sciences because the study received ethical clearance (IR.AJUMS.REC.1399.070) from

## Abstract

Due to the increasing population of the world, the presence of harmful compounds, especially phthalate esters (PAEs), are one of the important problems of environmental pollution. These compounds are known as carcinogenic compounds and Endocrine-disrupting chemicals (EDCs) for humans. In this study, the occurrence of PAEs and the evaluation of its ecological risks were carried out in the Persian Gulf. Water samples were collected from two industrial sites, a rural site and an urban site. Samples were analyzed using magnetic solid phase extraction (MSPE) and gas chromatography-mass spectrometry (GC/MS) technique to measure seven PAEs including Di(2-ethylhexyl) phthalate (DEHP), butyl benzyl phthalate (BBP), diethyl phthalate (DEP), dibutyl phthalate (DBP), Dimethyl phthalate (DMP), di-n-octyl phthalate (DNOP), and Di-iso-butyl phthalate (DIBP). The BBP was not detected in any of the samples. The total concentration of six PAEs (Σ6PAEs) ranged from 7.23 to 23.7 µg/L, with a mean concentration of 13.7µg/L. The potential ecological risk of each target PAEs was evaluated by using the risk quotient (RQ) method in seawater samples, and the relative results declined in the sequence of DEHP >DIBP > DBP > DEP > DMP in examined water samples. DEHP had a high risk to algae, crustaceans and fish at all sites. While DMP and DEP showed lower risk for all mentioned trophic levels. The results of this study will be helpful for the implementation of effective control measures and remedial strategies for PAEs pollution in the Persian Gulf.

## 1. Introduction

Phthalate esters (PAEs) are one of the most common materials category that are widely utilized in the chemical industry and plastic production [1]. PAEs are classified into two groups, PAEs

the Ethics Committee of the Ahvaz Jundishapur University of Medical Sciences (Iran) under project No. ETRC–9901. The Ahvaz Jundishapur University of Medical Sciences had no role in the study design, data collection, and analysis, the decision to publish, or the preparation of the manuscript.

**Competing interests:** he authors have declared that no competing interests exist.

with high and low molecular weight [2]. PAEs with high molecular weight, such as Di(2-ethylhexyl) phthalate (DEHP) and butyl benzyl phthalate (BBP) are mainly added to plastics as softening compounds to increase flexibility, transparency and extend life [3]. While PAEs with low molecular weight, such as diethyl phthalate (DEP) and dibutyl phthalate (DBP), are widely used as solvents and in the production of perfumes and cosmetic products [2, 4]. PAEs such as Dimethyl phthalate (DMP), DEP, DBP, BBP, DEHP, and di-n-octyl phthalate (DNOP) are endocrine disrupting chemicals) EDCs(, and the EPA has classified them as the priority pollutants [5]. Currently, the consumption of PAEs is several million tons per year, and the global consumption of PAEs in 2021 was about 3.6 million tons per year [6]. This wide use of PAEs, which is increasing every year, has caused PAEs to be found in different environmental elements, such as water, soil, and tissues of living organisms [7]. Despite the low vapor pressure, PAEs do not have a covalent bond to the polymer matrix and are not connected, and can be easily separated from plastic products and enter the environment [8]. Until now, various devices, such as high-performance liquid chromatography (HPLC), gas chromatography (GC) and GC coupled with mass spectrometry (GC-MS) have been used to identify PAEs. Also, in order to prepare the required conditions for identification of PAEs from several preparation methods such as single drop microextraction (SDME), liquid-liquid extraction (LLE), solvent extraction (SE) and liquid phase microextraction (LPME), rapid solvent extraction (ASE), stir bar absorption extraction (SBSE) and solid phase extraction (SPE) have been employed [9]. The problem with these preparation methods is their cost, time-consuming and high sensitivity [10]. Recently, magnetic solid phase extraction is used, which is a different type of solid-phase extraction (SPE). These adsorbents have magnetic properties and other methods, such as filtration and centrifugation are not required for preparation. Also, this method can facilitate the time and prevent column clogging [11, 12].

Humans are continuously exposed to PAEs through ingestion, inhalation and dermal adsorption, leading to adverse health effects [13, 14]. So far, acute toxicity caused by PAEs has not been reported, and their chronic toxicity depends on the type of their ester and is limited to laboratory research studies, including hepatotoxic [15], teratogenic, and carcinogenic effects [16]. Therefore, they are harmful to humans and the environment [17]. The results reported in 2009, showed that PAEs have a great effect on reducing the cases of pregnancy and fetal death [18]. The concentration of PAEs reported in the studies carried out in water environments around the world, have been measured at about 0.1 to 300 µg/L, which indicates the elevation of these compounds in the water phase [19, 20]. DEHP and DBP are the most common PAEs that have been detected in all environmental samples, including atmosphere, water, sediments, sludge, soil, and landfills [21]. Pollutions such as PAEs can also enter seawater through pathways, such as wastewater discharge, river runoff, and atmospheric deposition [22, 23]. Previous studies have shown the toxicity of PAEs to aquatic species at different trophic levels in algae, crustaceans, invertebrates and fish [6]. Yaru Cao and colleagues reported that DBP, Di-isobutyl phthalate (DIBP), and DEHP posed low to high potential risks to sea organisms at different trophic levels [6]. Ecological risk assessment is used to evaluate the risks caused by substances released by humans on living organisms in different ecosystems [24]. Ecological risk assessment is the process of estimating the probability of a certain event occurring under certain conditions. It is the basis for balancing and comparing the risks associated with environmental issues, and is a systematic way to ensure that risks are predicted and understood. The main goal of ecological risk assessment is ecological health at the entire ecosystem level [25, 26]. Therefore, the ecological risk assessment caused by the introduction of these compounds into the environment is essential for the implementation of the correct environmental management practices. Land-based anthropogenic activities are known as primary sources of pollutants, and the major distribution driving force of pollutants in oceans are oceanographic

geographical features of the area. Definitely, the concentrations of pollutants in water resources spatially oceans may mostly be related to the land-based human activities and the hydrological ecosystem dynamics of the coastal zone [27], so the land-based sources of pollution must be examined as an important factor affecting pollution concentration and distribution in water resources.

The Persian Gulf is one of the most sensitive and important marine environments, which is located in an area with latitude 24–30 N˚ and longitude 56–48 E˚ [28]. The existence of oil and gas resources and the high traffic of fuel and cargo ships, have made this bay a very strategic area but have also caused many environmental problems because of pollution release in this area [29]. Petroleum hydrocarbons and various heavy metals have been found in the water and sediments of these areas [30].

So far, a few studies have been conducted on PAEs measurement in the Persian Gulf. Considering the toxicity of PAEs and the importance of the Persian Gulf ecosystem and its sensitivity, this study aimed to evaluate the PAEs concentration in water samples and assess the potential ecological risk of PAEs in the Persian Gulf. To the best of our knowledge, this is the first study that evaluated PAEs contamination in different Persian Gulf sites based on land use, including industrial use, urban use, and rural use.

## 2. Methods and materials

### 2.1 Chemicals/reagents

For this study, all other chemicals/reagents used were DMP ($C_{10}H_{10}O_4$ -Purity $\geq$99%), DEP ($C_{12}H_{14}O_4$ -Purity $\geq$99%), DBP ($C_{16}H_{22}O_4$ -Purity $\geq$99%), BBP ($C_{16}H_{20}O_4$ -Purity $\geq$99%), DEHP ($C_{24}H_{38}O_4$ -Purity $\geq$99%), Dioctyl phthalate (DOP) ($C_{24}H_{38}O_4$ -Purity $\geq$99%), DIBP ($C_{16}H_{22}O_4$ -Purity $\geq$99%) and acetone ($C_3H_6O$ -Purity $\geq$99.5%) were purchased from Sigma-Aldrich. Also, Sulfuric acid ($H_2SO_4$ -Purity $\geq$95%), benzyl benzoate ($C_{14}H_{12}O_2$ -Purity $\geq$98%), n-hexane ($C_6H_{14}$ -Purity $\geq$95%) and dichloroethane ($C_2H_4Cl_2$ -Purity $\geq$99%), were all purchased from Merck, Germany.

### 2.2. Study area

Water sampling was performed aboard the Persian Gulf that is a semi-enclosed marginal sea. The Persian Gulf (Fig 1) has an average depth and maximum width of 36 m and 338 km respectively, located between 24˚ to 30˚ 30' N latitude and 48˚ to 56˚ 25' E longitude. As a result of regional hydrographic features and ongoing climatic changes/global warming, the Persian Gulf displayed the highest sea surface temperature records in the world. With increasing effluent discharges from neighboring countries, the Persian Gulf has experienced significant ecological stress, particularly in shallow coastal areas. The Persian Gulf region has also seen severe negative impacts imposed by seawater desalination, marine transportation, and power plants [31].

The sample collection points were determined by global positioning system (GPS), and the geographic information system (GIS) technique was used for zoning the concentration of PAEs to better display the pollution distribution in different environmental matrices. The geographic location of the sampling sites is shown in Fig 1. After the inspections were carried out in the target area, 4 different sites of the Persian Gulf region were chosen for water sampling; two sampling sites with industrial use in Asaluye (Fig 1D) and Mahshahr (Khor Musa) (Fig 1A), one sampling site with urban use (Bushehr) (Fig 1B), and one sampling site with rural use (Tangistan coast) (Fig 1C). Detailed sampling information, such as locations and regional descriptions of the investigated areas are shown in Table 1.

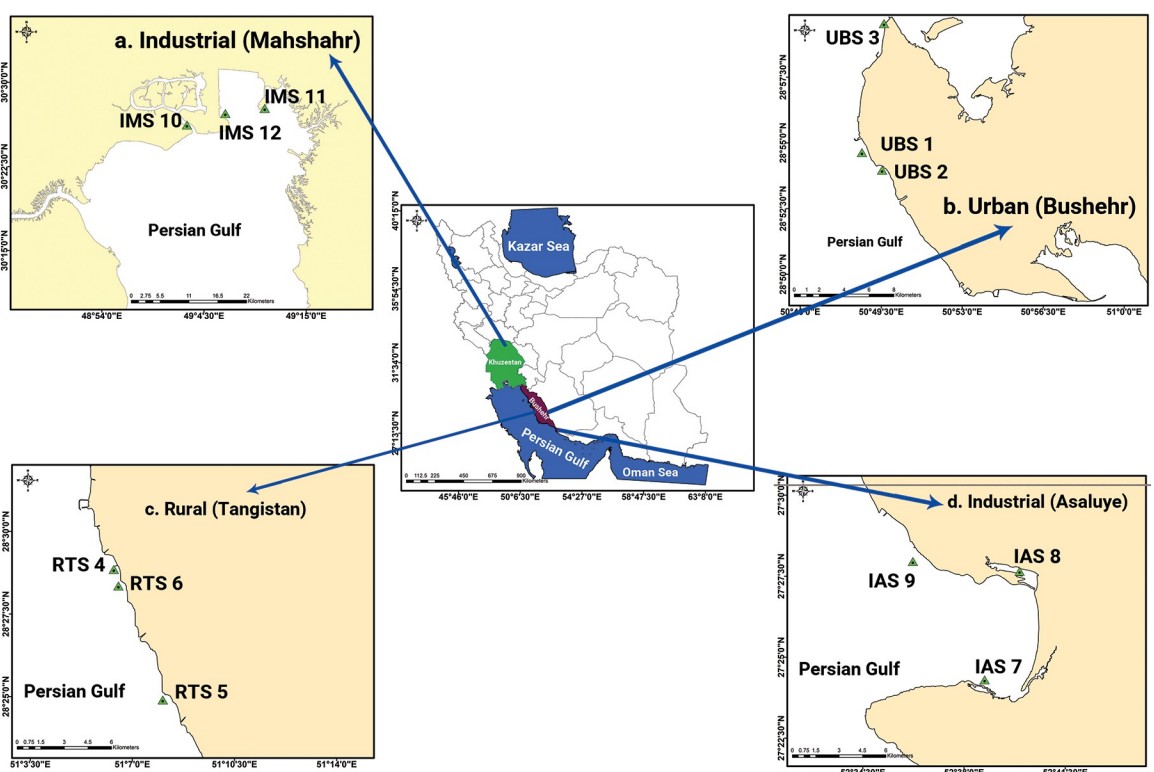

**Fig 1. Geographical location of the seawater sampling sites in Persian Gulf. a.** industrial use (Mahshahr area) (Khor Musa), **b**. urban use (Bushehr), **c.** rural use (Tangistan coast), **d.** industrial use (Asaluye).

## 2.3. Sample collection

A total of 36 seawater samples were collected over a period of six months at two-month intervals from three points of four introduced sampling sites in 2021. Field site access was granted by the Head Office of the Bushehr Province Environmental department. In each sampling, 1L of the seawater was collected by grab sampling so that the sample container submerges to a

**Table 1. Geographical location of the sea water sampling stations.**

| Sampling Point | | | UTM | |
|---|---|---|---|---|
| Name | | Code | X | Y |
| Urban (Bushehr) | Shoghab | UBS1 | 483148 | 3207146 |
| | Rey Shahr | UBS2 | 481587 | 3198085 |
| | TV | UBS3 | 483001 | 3196847 |
| Rural (Tangsten) | Khor Shahab | RTS4 | 510382 | 3150477 |
| | Kerry | RTS5 | 510641 | 3149548 |
| | Banjo | RTS6 | 513125 | 3143177 |
| Industrial (Asaloye) | Halo | IAS7 | 662651 | 3032369 |
| | Hara forest | IAS8 | 658475 | 3039065 |
| | Desalination | IAS9 | 664559 | 3038584 |
| Industrial (Mahshahr) | Khor Jafari | IMS10 | 325176 | 3371316 |
| | Petrochemical | IMS11 | 312334 | 3368785 |
| | Khor Smiley | IMS12 | 318679 | 3370581 |

depth of 0.5 m below the water surface to avoid surface scum and debris in the water. All of the samples were collected in clean Polytetrafluoroethylene (PTFE) bottles. Samples were stored at 4°C until analysis.

## 2.4. Sample preparation and analyses

In order to prepare the samples and extract the PAEs, carbon nanotubes magnetized by $Fe_3O_4$ were used as adsorbents [32, 33]. For this purpose, 10 mL of the sample was passed through a filter (0.7 μm). 0.5 g of salt, and 0.01 g of carbon nanotubes were added to it and shaken for 10 min. Then, using a magnet, the adsorbent was separated from the solution and dried at 50–60°C. Afterwards, 1 mL of n-hexane was added to the sample and shaken for 5 min. Finally, the adsorbent was separated using a magnet, and the samples were eventually ready for analysis.

The analysis of PAEs was performed using gas chromatography coupled with a mass spectrometer) GC-MS; 7890N, AGILENT & MS 5975C, MODE, Split/Spitless). The test conditions for PAEs analysis using GC-MS were as follows; column: DB5-MS column 30 m × 0.25 mm I. D × 0.25 μm film thickness; Inlet Temperature: 290°C; program temperature: Initial temperature 70°C for 1 min, Then the temperature increases at of 10°C /min up to 300°C and hold for 7 min; Flow Gas(carrier): high-purity helium (purity > 99.999%) with 1.0 mL/min (constant flow); Interface temperature: 290°C; Injection Type: Split less; and Inject Volume: 1 μl.

For making the standard curve, a series of standard solutions containing 5, 50, 100, 200, 500, 1000, and 2000 ng mL$^{-1}$ of PAEs were prepared, and by plotting the obtained peak areas versus the examined concentrations of each standard solution with triplicate measurements the calibration curves were manufactured. The aliphatic-terephthalate was applied as an internal standard. A sample of the output peak of the device is presented in Fig 2.

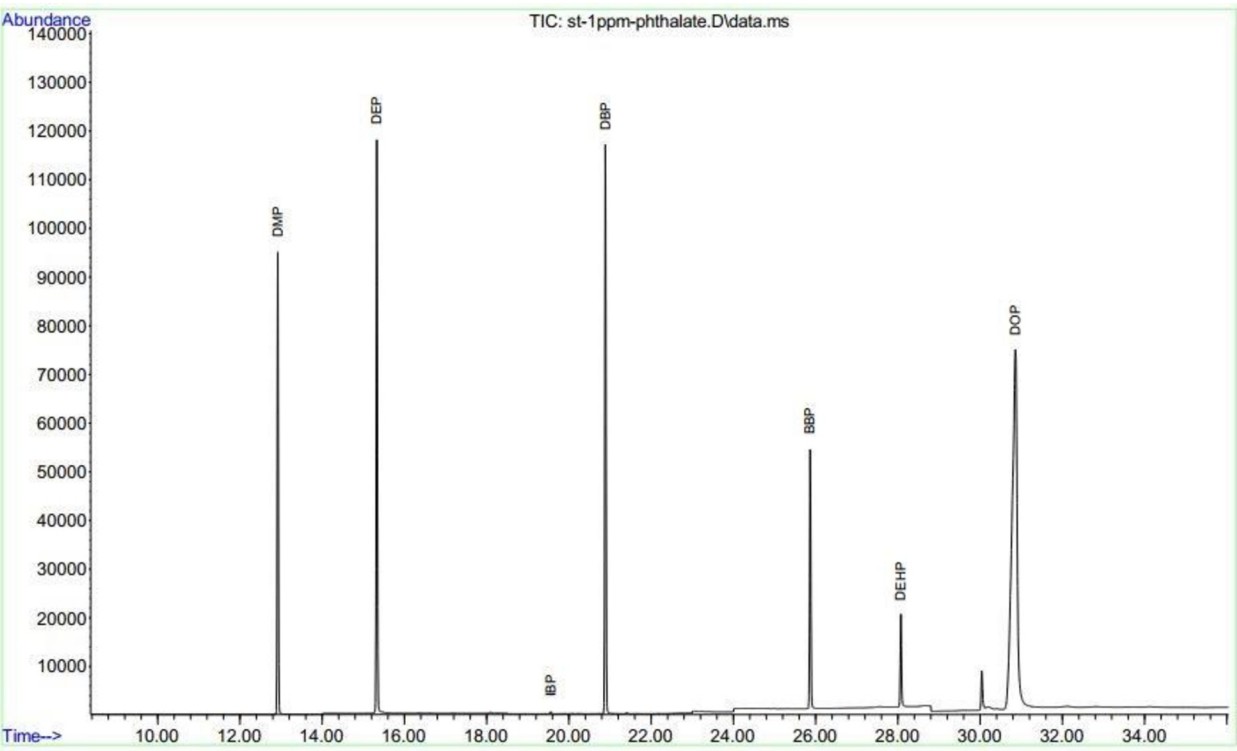

**Fig 2. Chromatogram plot of PAEs.**

## 2.5. Quality assurance and quality control

The GC-MS device were calibrated with calibration standards that were run with each batch of samples, and there was a good linearity in the detected range, and correlation coefficients (R2) were >0.992. the obtained values for R2, dynamic linear range (DLR), limit of detection (LOD) and limit of quantification (LOQ) of the seven targeted PAEs are presented in Table 2.

## 2.6. Potential ecological risk assessment

In order to calculate the ecological risk assessment, the Technical Guidance Document (TGD) of the European Commission was used [34]. The ecological risk of each PAE in seawater was calculated using the risk quotient (RQ) as follows:

$$RQ = MEC/PNEC \ (\mu g/L) \tag{1}$$

$$PNEC = (LC_{50} \ or \ EC_{50})/AF \tag{2}$$

Where, MEC and PNEC are the maximum measured concentration and the predicted no-effect concentration of the target PAE, respectively. Standard assessment factor (AF) for short-term/acute toxicity is 1000. Alternatively, PNEC values were also obtained by using long-term/chronic no effect concentration (NOEC) values for 1, 2, and 3 trophic levels, divided by an AF of 100, 50, and 10, respectively. Data on short-term or long-term toxicity of PAEs to aquatic organisms, including algae, crustaceans, and fish, were obtained from the USEPA ECOTOX database (http://cfpub.epa.gov/ecotox) and published literature.

If the RQ > 0.01, the pollutant has a low risk, if 0.01 <RQ< 1, the situation is medium risk, and if RQ> 1, the pollutant has a high risk [34, 35].

## 2.7. Data analysis

The Software Package for Social Sciences (SPSS) version 20.0 was used for statistical analysis of the obtained results. The Kolmogorov-Smirnov test was used to ensure that the data distribution was normal and then decided on the chosen suitable statistical test. The nonparametric Kruskal-Wallis and the one-way analysis of variance (ANOVA) tests followed by Tukey's post-hoc test were performed to compare the PAEs concentrations in different sampling sites. Also, we used the One-sample T-test presented for comparison of the mean contents of seawater samples with maximum permissible concentration. For this analysis, we used the standard value of 3 ug/L, which was presented in Okpara Kingsley's article [36]. A significant difference exists if the p-value is lower than 0.05.

**Table 2. Quality assurance/quality parameters for the extraction and analysis of Σ7PAEs.**

| PAEs | $R^2$ | LOD (µg/L) | LOQ (µg/kg) | DLR (µg/L) |
|------|-------|-----------|-------------|------------|
| DMP  | 0.993 | 1.68      | 5.56        | 5–1000     |
| DEP  | 0.995 | 0.860     | 2.84        | 5–1000     |
| DiBP | 0.997 | 0.810     | 2.62        | 5–1000     |
| DBP  | 0.990 | 0.750     | 2.46        | 5–1000     |
| DEHP | 0.991 | 0.100     | 0.340       | 5–1000     |
| BBP  | 0.992 | 1.67      | 5.50        | 5–1000     |
| DOP  | 0.996 | 2.11      | 6.97        | 5–1000     |

## 2.8. Ethics approval

The study received ethical clearance from the Ethics Committee of the Ahvaz Jundishapur University of Medical Sciences (Iran) under Project No. ETRC–9901.

## 3. Results and discussion

### 3.1. Occurrence of PAEs in seawater

The concentration of the targeted PAEs (DEHP, DIBP, DOP, DBP, DEP, DMP, BBP) were determined in examined seawater samples and reported in Table 3. All studied PAEs, except BBP, were detected at least in one seawater sample. In terms of the frequency of the presence of these compounds in examined water samples, the following order is established: DEHP > DIBP > DOP > DBP > DEP > DMP > BBP (Fig 3). The total concentration of six PAEs (Σ6PAEs) ranged from 7.23 to 23.7 μg/L, with a mean concentration of 13.7 μg/L. It is possible that discharged effluent from some small chemical industries, agricultural, and household wastes may have positive role on increased concentration of PAEs in the Persian Gulf [37]. DEHP, DOP, and DIBP were detected in all sampling sites (during three sampling times) with average concentrations of 11.3, 1.15 and 1.15 μg/L. The determined concentration of DEHP was higher than the values reported for the South China Sea (90.4 ng/L) [6]. Among the studied PAEs, DEHP revealed the highest abundance and concentration, which was consistent with the results reported for the water samples of Jiulong River in China [1]. The higher concentration of compounds such as DEHP and DBP may be related to their $K_{ow}$, as the higher the $K_{ow}$ for PAES compounds, the greater their lipophilic properties and as a result, they show resistance to chemical and biological degradation [7, 38]. Arun Elaiyaraja and colleagues determined the concentration of PAEs in water samples from rivers of southern India, and reported the mean concentration of PAEs as 35.6 μg/L, and DBP, DEHP, and DIBP were frequently detected in all water samples [39].

The results of the Kruskal Wallis test showed that the mean concentration of studied PAEs, except for BBP and DEHP, in different sampling sites were statistically significant (P<0.05). The results showed that the concentration of DMP, DEP, IBP and DBP in Mahshahr industrial site was significantly higher than other areas. Also, the post-hoc analysis of IBP concentration revealed that the mean concentration of IBP was significantly higher in Mahshahr than

**Table 3. The average concentration of PAEs μg/l in seawater samples.**

| Studied sites | Sampling point | DMP | DEP | DIBP | DBP | BBP | DEHP | DOP |
|---|---|---|---|---|---|---|---|---|
| Urban (Bushehr) | UBS1 | ND | ND | 2.51 | ND | ND | 15.8 | 1.72 |
| | UBS2 | ND | ND | 0.512 | ND | ND | 11.7 | 1.70 |
| | UBS3 | ND | ND | 0.701 | ND | ND | 13.5 | 4.61 |
| Rural (Tangsten) | RTS4 | ND | ND | 0.410 | ND | ND | 6.21 | 0.612 |
| | RTS5 | ND | ND | 0.513 | ND | ND | 6.90 | 0.201 |
| | RTS6 | ND | ND | 0.311 | ND | ND | 5.71 | 0.400 |
| Industrial (Asaloye) | IAS7 | ND | ND | 1.82 | ND | ND | 12.3 | 0.501 |
| | IAS8 | ND | ND | 0.201 | ND | ND | 13.1 | 0.310 |
| | IAS9 | ND | ND | 1.81 | ND | ND | 18.4 | 0.402 |
| Industrial (Mahshahr) | IMS10 | ND | ND | 1.00 | ND | ND | 6.62 | 0.501 |
| | IMS11 | ND | 0.200 | 1.00 | 0.110 | ND | 7.93 | 0.511 |
| | IMS12 | 0.101 | 0.212 | 3.11 | 0.400 | ND | 17.51 | 2.40 |
| Max | - | 0.201 | 0.300 | 3.31 | 0.511 | ND | 18.51 | 5.01 |
| Min | - | ND | ND | ND | ND | ND | ND | ND |

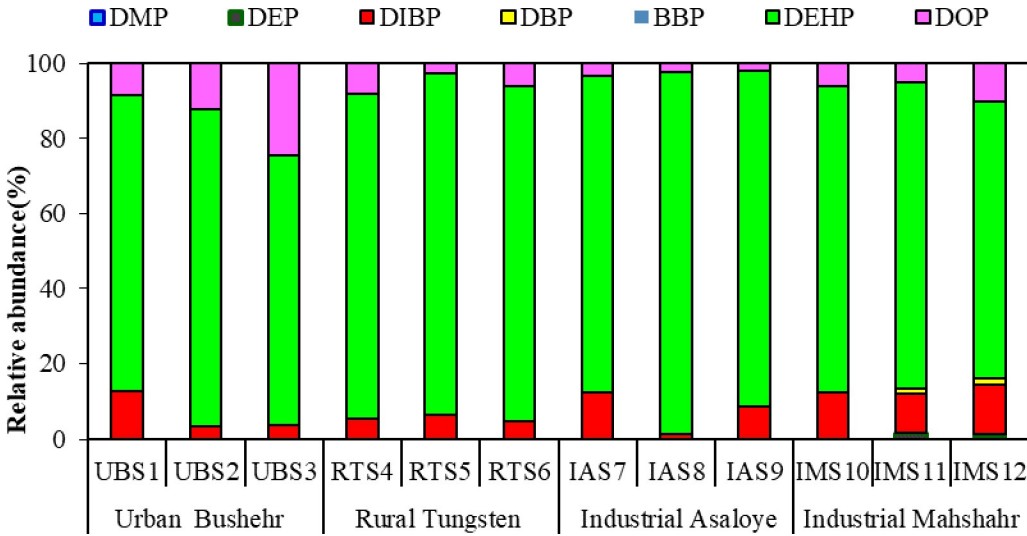

**Fig 3. Frequency of presence of the studied PAEs in the sea water samples at different sampling sites.**

Asaluye and Tangestan. However, there was no significant difference between Mahshahr and Bushehr sampling point. The mean concentration of DOP in Bushehr was significantly higher than Asalouye and Tangestan, but the difference between Bushehr and Mahshahr was not significant.

The results of the one-sample wilcoxon signed-rank test showed a significant difference between the mean PAEs in all four regions under review with the standard USEPA value (P<0.001). According to this analysis, the largest difference in PAEs means was related to Bushehr region, which was statistically significant (P = 0.007). The mean of PAEs in the Tangestan region was 7.00±0.70 μg/L, and the difference with the USEPA value was 4.00 μg/L, which indicated the lowest mean among the evaluated regions (P = 0.007). Furthermore, the mean PAE levels in Asaluyeh and Mahshahr were 16.44±3.46 and 13.78±7.46 μg/L, respectively, and the difference between these areas and USEPA level was statistically significant (P<0.05). Finally, the mean total PAE level in these four regions was reported as 13.72±5.91 μg/L and its difference with USEPA level was 10.72 μg/L (P<0.001) (Table 4). In line with our results, Kingsley et al. showed that the total mean concentration of PAEs was higher than the criteria of 3 ug/L for PAEs suggested for the protection of fish by the United States Environmental Protection Agency (USEPA) [36]. Okpara et al believe that PAEs are endocrine disrupting chemicals whose effects may alter the function of the endocrine system in humans [40]. These findings emphasize the evaluation of PAEs in different regions due to the environmental exposure of humans to these compounds and their carcinogenic risk.

### 3.2. Distribution of PAEs in seawater

Fig 4 shows the distribution of PAES in different sampling points. The highest DEHP concentration was 18.4 μg/L, detected at the IAS9 sampling point. The highest detected concentrations of DOP and DIBP were 4.61 and 3.11 μg/L that were observed in samples taken from UBS3 and IMS12 points, respectively. DMP and DEP were detected only in samples collected from Mahshahr industrial site with an average concentration of 0.101 and 0.206 μg/L. Among the sites, Bushehr urban site (UBS1, UBS2, UBS3) and Asaloye Industrial site (IAS7, IAS8, IAS9) had the highest concentration of Σ6PAEs with a total of 52.7 μg/L and 48.8 μg/L, respectively. This results may be due to the discharge of wastewater and household waste into

**Table 4. Comparison of the mean PAE among regions.**

| Region | PAE | | |
|---|---|---|---|
| | Mean ± SD | Mean difference | p-value* |
| **Bushehr** | 17.67±2.82 | 14.66 | 0.007 |
| **Tangestan** | 7.00±0.70 | 4.00 | 0.007 |
| **Asaloyeh** | 16.44±3.46 | 13.44 | <0.001 |
| **Mahshahr** | 13.78±7.46 | 10.77 | 0.003 |
| **ΣPAE** | 13.72±5.91 | 10.72 | <0.001 |

*One-sample Wilcoxon Signed-Rank Test

seawater in industrial (Asaloye) sites [41, 42]. Similar studies reported that the highest concentration of PAEs detected in area near the urban areas because of the existence of extensive industrial and commercial activities. In these areas, many industrial and domestic effluent are discharged into the water bodies [36]. In contrast, the rural sites of Tangistan (RTS4, RTS 5, RTS 6) had the lowest amount of Σ6PAEs with an average concentration of 21.2 μg/L. In rural areas, the concentration of PAEs was lower, because in these areas fewer industrial and domestic wastewaters discharge into the water resources, which is consistent with previous studies [4, 19].

### 3.3. Comparison of the chemical composition of PAEs in other locations

The investigations that have been carried out so far have mostly been conducted on compounds such as DBP, DEHP, DMP, DEP and DIBP, and there are limited data on DOP and BBP. In order to better understanding the state of PAEs pollution in water resources, the concentrations reported by different studies are shown in S1 Table. In the present study, DEHP

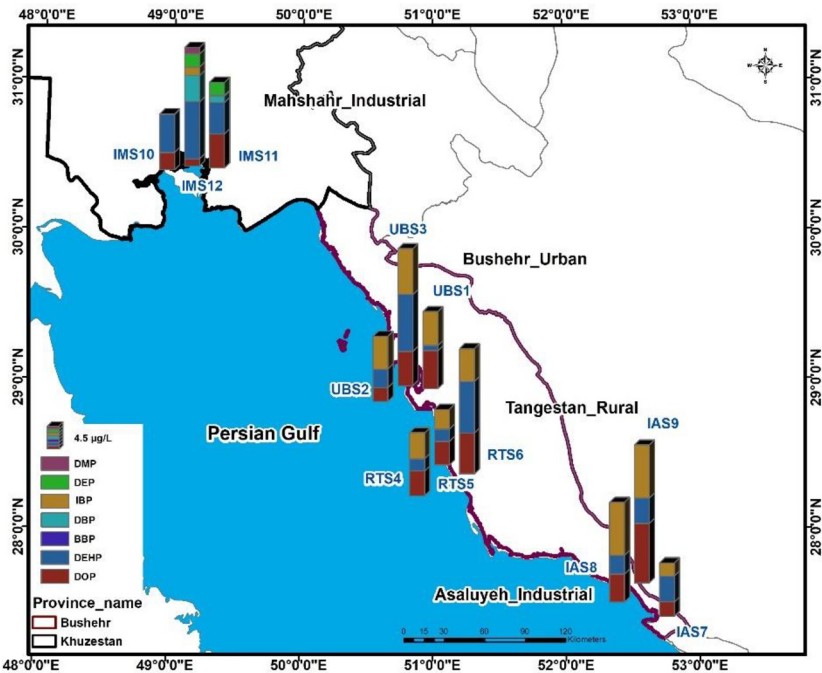

**Fig 4. Distribution of Σ7PAEs in seawater.**

and DIBP were the most abundant compounds detected in all examined water samples. The highest concentration of PAEs is related to DEHP, whose concentration in the study area is similar to some areas in the world [43]. The concentration of this pollutant in Jiulong and Songhua Rivers in China is also high, which is probably related to the existence of more and continuous human activities in these areas [1, 44]. In contrast, BBP was the only compound that was not detected in most studies in the aquatic environment [45]. In Songhua and Yangtze Rivers in China, the concentrations of DEP, DMP and DBP are about 10 to 20 times higher than the concentrations detected in the Persian Gulf. It can be due to the discharge of domestic, industrial and agricultural wastewater into the aquatic environments along the Songhua and Yangtze Rivers in China [44, 46]. In another study that was conducted to measure the concentration of PAEs in drinking water bottles in Tehran, the results showed that among PAEs, DEHP has the highest concentration (2.22 μg/L) [9]. In the study of Kingsley et al., six types of phthalate esters (PAEs) were analyzed in water samples of Surface Water of U-Tapao Canal. However, only three types of PAEs, including DBP, DEHP, and DiNP, were identified. The total concentration of PAEs in this study varied between 1.44 and 12.08 μg/L and DiNP and DEHP posed a high risk to algae, crustacean and fish. They also reported higher levels of PAEs for sampling sites that are located near commercial and industrial areas that may be related to the discharging industrial effluents and domestic wastewater into the canal at these examine points [36].

### 3.4. Ecological risk assessment

All aquatic organisms living in aquatic environments are inevitably exposed to various pollutants, such as PAEs [36]. In the present study, the potential risks of PAEs in the Persian Gulf were evaluated on living organisms including algae, crustaceans and fish. To calculate the risk, acute or chronic toxicity (LC50, EC50, and NOEC) data for three trophic levels of the environment were employed [47]. Table 5 shows the toxicity data for Σ5PAEs (DEHP, DIBP, DMP, DEP, DBP).

Fig 5 revealed the results of calculated RQ for studied PAEs. The RQ values for five PAEs in seawater samples were as DEHP > DIBP > DBP > DEP > DMP. Based on these results, the RQ value for DEHP is higher than 1 in all sampling point except in RTS6, and it shows that DEHP may have a high risk for algae, crustaceans and fish. But in RTS6, DEHP has medium risk to fish, and RQ value is between 1 and 0.01. Previous studies showed that PAEs such as

**Table 5. Toxicity of PAEs in some sensitive aquatic organisms.**

| PAEs | Population | Species | Toxicity data (μg/L) | AF | PNEC (μg/L) |
|---|---|---|---|---|---|
| DEHP | Algae | Pseudokirchneriella subcapitata | 96 h, population, EC50 = 100 | 1000 | 0.1 |
| | Crustaceans | Mytilus edulis | 21 d, mortality, NOEC = 42 | 50 | 0.84 |
| | Fish | Gasterosteus aculeatus | 28 d, mortality, NOEC = 300 | 50 | 6 |
| DMP | Algae | Pseudokirchneriella subcapitata | 96 h, population, NOEC = 10,000 | 10 | 1000 |
| | Crustaceans | Daphnia magna | 21 d, mortality, NOEC = 9600 | 10 | 960 |
| | Fish | Oncorhynchus mykiss | 102 d, mortality, NOEC = 11,000 | 10 | 1100 |
| DEP | Algae | Pseudokirchneriella subcapitata | 96 h, population, NOEC = 8106 | 10 | 810.6 |
| | Crustaceans | Americamysis bahia | 21 d, mortality, NOEC = 2700 | 10 | 270 |
| | Fish | Lepomis macrochirus | 28 d, morphological, NOEC = 1650 | 10 | 165 |
| DBP | Algae | Pseudokirchneriella subcapitata | 96 h, population, NOEC = 210 | 10 | 21 |
| | Crustaceans | Americamysis bahia | 21 d, mortality, NOEC = 260 | 10 | 26 |
| | Fish | Oncorhynchus mykiss | 99 d, growth, NOEC = 100 | 10 | 10 |
| DIBP | Algae | Pseudokirchneriella subcapitata | 96 h, population, NOEC = 210 | 10 | 21 |
| | Fish | Pimephales promelas | 96 h, mortality, LC50 = 900 | 1000 | 0.9 |

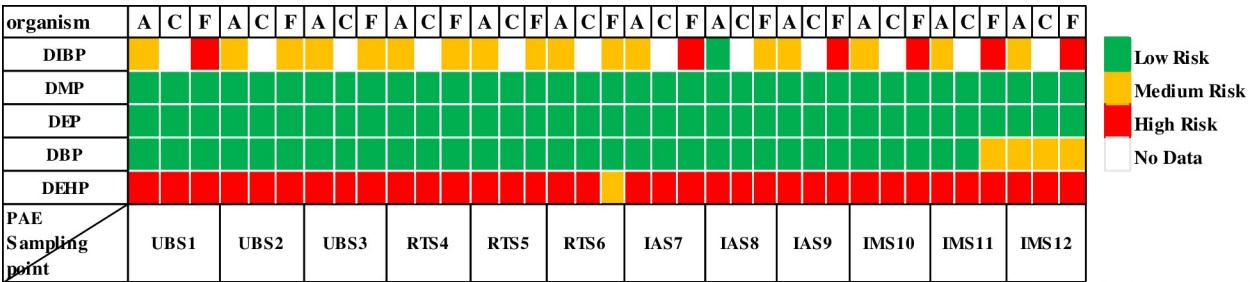

**Fig 5. Ecological risk assessment of PAEs for the aquatic organisms based on the calculated risk quotients (A, C, and F represent algae, crustaceans, and fish, respectively).**

DBP, DEHP, and DEP have endocrine-disrupting effects, growth, reproduction, and lifespan of Daphnia magna [48]. RQ for DEP and DMP were less than 0.01 at all sites, indicating that these two compounds are unlikely to pose a risk to algae, crustaceans and fish. The results of a study conducted in seawater in northern China demonstrated that in most sampling sites, DMP and DEP are unlikely to cause damage to aquatic organisms [6]. The RQ values for IBP for two trophic levels of algae and fish show that its value in most sampling points was 0.01<RQ<1 and displayed medium risk. In the sampling points of UBS1, IAS7, IAS9, IMS11 and IMS12, the RQ value for fish was higher than 1, with a higher risk. The RQ values for DBP were lower than for DIBP, and at point IMS12 it had medium risk to algae, crustaceans and fish. It should be noted that DBP is gradually replaced by DIBP [49]. Hou and colleagues collected the water samples from five functional areas of Baiyangdian including the primitive area, tourism area, living area, breeding area, and inflow area, and analyzed for concentrations of PAEs. The order of detected ∑PAEs concentration in studied areas was as: inflow area->aquaculture, living areas, and tourist areas>primitive area, and higher concentrations are related to the influence of industrial sewage, human activities, and pesticides and fertilizers usage [50]. Wang and his colleagues found that the risk assessment of seawater indicated that DEHP and DiBP may pose potential high risks for sensitive organisms, and DnBP may indicate moderate ecological risks [51].

## 5. Conclusions

This study provides an overview of the concentrations of PAEs and their distribution in the Persian Gulf and also assessed the potential ecological risks that these compounds may pose to the environment. All PAEs, except BBP, were detected in all examined water samples with detection frequencies, ranging from 0.4 to 96.3%. In terms of the frequency of the presence of these compounds in examined water samples, the following order is established as follows: DEHP > DIBP > DOP > DBP > DEP > DMP > BBP. The distribution of PAEs in the vicinity of the Persian Gulf showed that urbanization had significant effect on the concentration of PAEs. The mean concentration of the PAEs in the urban area was 52.7 μg/L that was higher than other areas. The ecological risk assessment of PAEs in Persian Gulf seawater showed that DEHP and DIBP have the highest potential risk in the aquatic environment, while DEP and DMP have the lowest risk. Therefore, this issue is a serious concern that should be resolved by making effective management decisions.

## 6. Study strengths and limitations

This study is one of the few studies that investigated the concentration of some emerging pollutants in different areas of the coasts of the Persian Gulf with various land uses. The

information obtained from this study will be very useful in matters related to environment and aquatic organisms' protection, and the risk management of PAEs and planning for sustainable development.

However, the present study has limitations such as the limited number of examined samples and studied areas. Undoubtedly, further sampling of multiple areas over a long period of time can reveal the long-term impact of industrial development on the quality of coastal waters in that area. Also, oceanographic geographical features are another possible factor associated with distribution of pollution, which has been ignored in this study. In this study, it was not possible due to both financial and time limitations. Direct hunting is suggested to monitor contaminants such as PAEs in the food chain and aquatic organisms.

## Supporting information

**S1 Table. Comparison of PAEs concentrations in global water resources (μg/L).**
(DOCX)

**S1 Graphical abstract.**
(TIF)

## Author Contributions

**Conceptualization:** Ali Akbar Babaei.

**Data curation:** Maria Khishdost, Afshin Takdastan, Ali Akbar Babaei.

**Investigation:** Maria Khishdost, Afshin Takdastan, Ali Akbar Babaei.

**Methodology:** Sina Dobaradaran, Gholamreza Goudarzi, Afshin Takdastan, Ali Akbar Babaei.

**Writing – original draft:** Maria Khishdost.

**Writing – review & editing:** Sina Dobaradaran, Gholamreza Goudarzi, Afshin Takdastan, Ali Akbar Babaei.

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
