## [Decision Letter · Decision Letter 0]

27 Mar 2023

PONE-D-23-03668

Contaminant occurrence, distribution and ecological risk assessment of phthalate esters in the Persian Gulf

PLOS ONE

Dear Dr. Babaei,

Thank you for submitting your manuscript to PLOS ONE. The review of your manuscript, Contaminant occurrence, distribution and ecological risk assessment of phthalate esters in the Persian Gulf, submitted to PLOS ONE has been completed. The Reviewers recommend the need for major revisions. Therefore, we invite you to submit a revised version of the manuscript that addresses the points raised during the review process.

We look forward to receiving your revised manuscript.

Kind regards,

Hai O. Xu

Academic Editor

PLOS ONE

Journal Requirements:

   "This study is the result of the Ph.D. thesis in Environmental Health Engineering of Ahvaz Jundishapur University of Medical Sciences of Medical Sciences (No. ETRC‒9901)."

    "We thank Ahvaz Jundishapur University of Medical Sciences because the study received ethical clearance (IR.AJUMS.REC.1399.070) from the Ethics Committee of the Ahvaz Jundishapur University of Medical Sciences (Iran) under project No.  ETRC‒9901. The Ahvaz Jundishapur University of Medical Sciences had no role in the study design, data collection, and analysis, the decision to publish, or the preparation of the manuscript."

7. We note that Figures 1 and 3 in your submission contain [map/satellite] images which may be copyrighted. All PLOS content is published under the Creative Commons Attribution License (CC BY 4.0), which means that the manuscript, images, and Supporting Information files will be freely available online, and any third party is permitted to access, download, copy, distribute, and use these materials in any way, even commercially, with proper attribution. For these reasons, we cannot publish previously copyrighted maps or satellite images created using proprietary data, such as Google software (Google Maps, Street View, and Earth). For more information, see our copyright guidelines: http://journals.plos.org/plosone/s/licenses-and-copyright.

a. You may seek permission from the original copyright holder of Figures 1 and 3 to publish the content specifically under the CC BY 4.0 license.  

Additional Editor Comments:

One or more of the reviewers has recommended that you cite specific previously published works. Members of the editorial team have determined that the works referenced are not directly related to the submitted manuscript. As such, please note that it is not necessary or expected to cite the works requested by the reviewers.

Reviewers' comments:

Reviewer's Responses to Questions

**Comments to the Author**

1. Is the manuscript technically sound, and do the data support the conclusions?

Reviewer #1: Partly

Reviewer #2: Yes

Reviewer #3: Yes

2. Has the statistical analysis been performed appropriately and rigorously? 

Reviewer #1: Yes

Reviewer #2: Yes

Reviewer #3: Yes

3. Have the authors made all data underlying the findings in their manuscript fully available?

Reviewer #1: Yes

Reviewer #2: Yes

Reviewer #3: Yes

4. Is the manuscript presented in an intelligible fashion and written in standard English?

Reviewer #1: No

Reviewer #2: Yes

Reviewer #3: Yes

5. Review Comments to the Author

Reviewer #1: - The paper has some typographical and grammatical errors which must be corrected. Therefore, the English language of the text should be strongly revised by a native English speaker with expertise in the scientific field and skills in scientific paper writing.

- The "conclusion" section in the abstract part must be improved.

- The novelty is not sufficiently explained or clears in the introduction section. Also, the researches gap should be clearly described.

- The quality control/assurance and the analytical recovery data from PAEs in seawater samples must be presented.

- Calibration ranges for the considered analytes, as well as at least one chromatogram illustrating the separation of standard PAEs + separation of PAEs in a seawater sample must be provided.

- The results of the 'One sample T test' must be presented for comparison of the mean contents of in seawater samples with maximum permissible concentration.

- Quality of the discussion section must be improved. In so doing, the authors must be organized the discussion from the general to the specific, linking your findings to the literature, then to theory, then practice and avoid repetition from the introduction.

- Limitations of the study must be presented in the conclusion section.

- For numbers in text and tables < 1.00, use three digits beyond the decimal point; for numbers between 1.00 and 9.99 use two digits beyond the decimal point; for numbers between 10.0 and 99.9, use one digit beyond the decimal point; and for concentrations ≥ 100, use the nearest whole number.

- The following relevant reference must be cited for in-text citation:

Environmental Science and Pollution Research, 28(43): 61151-61162 (2021).

Reviewer #2: It is a good and practical study but I made the following observations which should be addressed by the authors. After doing the following points, it can be accepted in the Journal of PLOS ONE.

Abstract

- It needs to be rewritten. The abstract should be improve, very briefly the purpose of the study, methodology, results and conclusions should be mention.

Keywords

- Sort by alphabetical order.

- It is better to use keywords other than the title.

Introduction

- It is necessary to give explanations about the preparation method and equipment such as GC-MS, etc.

- It is necessary to explain a little about the risk assessment.

-In the introduction section, it is necessary to mention the limitations of the previous studies (briefly) and explain the reason for the current study in more detail. It is also necessary to briefly mention the benefits of doing this research.

- You can use the following articles to complete this section:

https://doi.org/10.1007/s11356-023-25313-0, https://doi.org/10.1080/03067319.2022.2062239, https://doi.org/10.1007/s11356-018-1471-y, https://doi.org/10.1007/s11356-021-14290-x,

Materials and methods

- Page 5, Line 109. Change to “(C₁₀H₁₀O₄ -Purity ≥99%),”

- Page 5, Line 113. Change to “(H₂SO₄ -Purity ≥95%),”

- The space between numbers and units should be observed throughout the text (such as Line 135. “1 L”, etc.)

- Line 140. It is necessary to add a reference or a summary of it to the magnetic absorber section.

- It is better to move the “Data analysis” part to the end of the materials and methods section.

- It is necessary to mention the preparation steps of the standard and internal standard in detail.

- It is necessary to mention the LOD, LOQ, Recovery and QC samples.

Result and Discussion

- The discussion part of the article is extremely weak, it is necessary to mention possible reasons for all the results and to compare with other works. Such as Part 3.1., 3.2., 3.3. and 3.4. Should be written in more detail (more comparisons should be made with past studies (at least 7 similar studies, even with other similar food matrix such as bottled water, etc.) and possible reasons for the higher or lower values of these compounds should be mentioned). You can use the mentioned articles.

- In Table 2, the maximum and minimum of each data should also be mentioned.

Conclusion

-Finally, state your limitations of the study and possible suggestions for other researchers.

Reviewer #3: （1） line 75: please specify whether the data about the PAEs is Persian Gulf or global

（2） Please supplement the existing research progress and data on PAEs.

（3） Line 184: Data on PAEs in the area before previous industrial, agricultural and domestic wastewater discharges are lacking, so it cannot be directly concluded that high concentrations of PAEs in the area originate from them. The specific type of industries in its vicinity should be given, not all of them can lead to PAEs pollution

（4） Line 187: Please give a specific value, how high is it?

（5） Line 188: please check the correctness of writing “PAES”.

（6） Line 209-Line 213: From the results and geographical location, the concentrations of Σ6PAEs in Bushehr urban site is higher than that in Asaloye Industrial site. Give a reasonable explanation for this.

（7） Line 234: please check the correctness of writing “Y angtze”

This is an intentional and interesting study. it provides an overview of the concentrations of PAEs and their distribution in the Persian Gulf and also assessed the potential ecological risks that these compounds may pose to the environment. However, the adequacy of the data, the reasonableness of the analysis and the Innovation of the study are still lacking. I hope you will make changes in response to the above suggestions

6. PLOS authors have the option to publish the peer review history of their article (what does this mean?). If published, this will include your full peer review and any attached files.

Reviewer #1: No

Reviewer #2: No

Reviewer #3: No

---

## [Author Response · Author response to Decision Letter 0]

27 May 2023

Editor’s and Reviewer’s comments:

Editor's comments:

Thank you a lot, for your advice. We checked it.

Field site access was granted by the Head Office of the Bushehr Province Environmental. We added this to the Method section.

 "This study is the result of the Ph.D. thesis in Environmental Health Engineering of Ahvaz Jundishapur University of Medical Sciences of Medical Sciences (No. ETRC‒9901)."

 "We thank Ahvaz Jundishapur University of Medical Sciences because the study received ethical clearance (IR.AJUMS.REC.1399.070) from the Ethics Committee of the Ahvaz Jundishapur University of Medical Sciences (Iran) under project No. ETRC‒9901. The Ahvaz Jundishapur University of Medical Sciences had no role in the study design, data collection, and analysis, the decision to publish, or the preparation of the manuscript."

Thanks for your nice overview. We removed the part Acknowledgments Section.

This part was edited as “All relevant data are within the manuscript.”

Many thanks, it was done in title page.

Many thanks, it was done.

 7. We note that Figures 1 and 3 in your submission contain [map/satellite] images which may be copyrighted. All PLOS content is published under the Creative Commons Attribution License (CC BY 4.0), which means that the manuscript, images, and Supporting Information files will be freely available online, and any third party is permitted to access, download, copy, distribute, and use these materials in any way, even commercially, with proper attribution. For these reasons, we cannot publish previously copyrighted maps or satellite images created using proprietary data, such as Google software (Google Maps, Street View, and Earth). For more information, see our copyright guidelines: http://journals.plos.org/plosone/s/licenses-and-copyright.

 a. You may seek permission from the original copyright holder of Figures 1 and 3 to publish the content specifically under the CC BY 4.0 license. 

Thanks for your attention, the figs changed, we used the https://data.humdata.org/organization/ocha-romena site

---

## [Editor Report · Decision Letter 1]

7 Jun 2023

Contaminant occurrence, distribution and ecological risk assessment of phthalate esters in the Persian Gulf

PONE-D-23-03668R1

Dear Dr. Babaei,

We’re pleased to inform you that your manuscript has been judged scientifically suitable for publication and will be formally accepted for publication once it meets all outstanding technical requirements.

Kind regards,

Hai O. Xu

Academic Editor

PLOS ONE